# Kaempferol Inhibits Zearalenone-Induced Oxidative Stress and Apoptosis via the PI3K/Akt-Mediated Nrf2 Signaling Pathway: In Vitro and In Vivo Studies

**DOI:** 10.3390/ijms22010217

**Published:** 2020-12-28

**Authors:** Peramaiyan Rajendran, Rebai Ben Ammar, Fatma J. Al-Saeedi, Maged E. Mohamed, Medhat A. ElNaggar, Saeed Y. Al-Ramadan, Gamal M. Bekhet, Ahmed M. Soliman

**Affiliations:** 1Department of Biological Sciences, College of Science, King Faisal University, Al-Ahsa 31982, Saudi Arabia; rbenammar@kfu.edu.sa (R.B.A.); gbekhet@kfu.edu.sa (G.M.B.); 2Laboratory of Aromatic and Medicinal Plants, Center of Biotechnology, Technopole of Borj-Cedria, Hammam-Lif 901 2050, Tunisia; 3Department of Nuclear Medicine, Faculty of Medicine, Kuwait University, Safat 13110, Kuwait; fatma.alsaeedi@ku.edu.kw; 4Pharmaceutical Sciences Department, College of Clinical Pharmacy, King Faisal University, Al-Ahsa 31982, Saudi Arabia; memohamed@kfu.edu.sa; 5Pharmacognosy Department, College of Pharmacy, Zagazig University, Zagazig 44519, Egypt; 6Plant Pathology Research Institute, Agricultural Research Center, Giza Governorate 12619, Egypt; drnaggar@sago.gov.sa; 7Research Central Laboratory, Saudi Grains Organization, Riyadh 11471, Saudi Arabia; 8Department of Anatomy, College of Veterinary Medicine, King Faisal University, Al-Ahsa 31982, Saudi Arabia; salramadan@kfu.edu.sa; 9Department of Zoology, Faculty of Science, Alexandria University Egypt, Alexandria 21544, Egypt; 10Department of Arid Land Agriculture, College of Agricultural & Food Sciences, King Faisal University, Al Ahsa 31982, Saudi Arabia; amohamed@kfu.edu.sa; 11Virus & Phytoplasma Research Department, Plant Pathology Research Institute, Agricultural Research Center, Giza Governorate 1266, Egypt

**Keywords:** zearalenone, kaempferol, PI3K/Akt, hepatotoxicity, Nrf2, apoptosis

## Abstract

In this study, kaempferol (KFL) shows hepatoprotective activity against zearalenone (ZEA)-induced oxidative stress and its underlying mechanisms in in vitro and in vivo models were investigated. Oxidative stress plays a critical role in the pathophysiology of various hepatic ailments and is normally regulated by reactive oxygen species (ROS). ZEA is a mycotoxin known to exert toxicity via inflammation and ROS accumulation. This study aims to explore the protective role of KFL against ZEA-triggered hepatic injury via the PI3K/Akt-regulated Nrf2 pathway. KFL augmented the phosphorylation of PI3K and Akt, which may stimulate antioxidative and antiapoptotic signaling in hepatic cells. KFL upregulated Nrf2 phosphorylation and the expression of antioxidant genes HO-1 and NQO-1 in a dose-dependent manner under ZEA-induced oxidative stress. Nrf2 knockdown via small-interfering RNA (siRNA) inhibited the KFL-mediated defence against ZEA-induced hepatotoxicity. In vivo studies showed that KFL decreased inflammation and lipid peroxidation and increased H_2_O_2_ scavenging and biochemical marker enzyme expression. KFL was able to normalize the expression of liver antioxidant enzymes SOD, CAT and GSH and showed a protective effect against ZEA-induced pathophysiology in the livers of mice. These outcomes demonstrate that KFL possesses notable hepatoprotective roles against ZEA-induced damage in vivo and in vitro. These protective properties of KFL may occur through the stimulation of Nrf2/HO-1 cascades and PI3K/Akt signaling.

## 1. Introduction

Fungal mycotoxins are produced naturally and infect a wide range of farm produce. Since they are specific to food products and are highly likely to impact human health and livestock and poultry production, mycotoxins are harmful to animals and humans worldwide [1,2]. Zearalenone (ZEA) is an oestrogenic mycotoxin responsible for polluting cereal grains, for instance oats, barley or sorghum across the globe. Many studies have shown that ZEA and its metabolites may be ingested and lead to various harmful effects on reproductive system, hepatocytes and kidney cells [3,4,5]. When ZEA enters the body of either animals or human beings, it passes through the hepatic and systemic vessels before reaching another target organ [3]. The liver has a significant role in carrying out detoxification of foreign substances along with elimination of metabolites. Reactive metabolites produce ROS during the biotransformation of foreign substances. Numerous chronic diseases are associated with ROS. Liver cirrhosis represents a condition in which oxidation and oxidation resistance are clearly imbalanced [6,7]. ROS are induced through the phosphoinositide 3-kinase (PI3K)/Akt pathway, which triggers various intracellular responses [8,9,10]. Hence, PI3K/Akt pathway significantly helps in providing cellular defense against inflammatory stimuli. Numerous studies have shown that multiple signal transduction mechanisms, such as PI3K/Akt, make Nrf2 dissociate from Keap1 and facilitate subsequent signal transduction to induce the activation of antioxidant enzymes [11,12,13], and also have demonstrated that continuous oxidative stress leads to the downregulation of cell survival PI3K/Akt signaling [14,15,16]. The regulation of PI3K/Akt signaling routes may therefore be a promising strategy for preventing ROS-induced hepatic apoptosis.

On other hand, numerous previous reports have demonstrated that natural compounds have several properties in inhibiting hepatic apoptosis. Flavonoids are antioxidants that are present in a wide range of plants and are abundant in nature. Kaempferol (3,5,7-trihydroxy-2-(4-hydroxyphenyl)-4H-1-benzopyran-4-one) represents the maximum encountered aglycone flavonoids. Its antioxidant and anti-inflammatory properties have been illustrated in numerous ailments such as diabetes, encephalomyelitis, carcinogenesis and asthma. Furthermore, KFL can effectively scavenge free radicals and preserve the functions of antioxidants [17,18]. This research aims to investigate the impact of KFL on ZEA-induced oxidative stress and apoptosis via PI3K/Akt pathway both in vitro and in vivo to elucidate the fundamental molecular mechanisms.

## 2. Results

### 2.1. KFL Inhibits Hepatotoxicity Induced by ZEA in HepG2 Cells

HepG2 cells were added to recommended amounts of ZEA for 24 and 48 h, as depicted in Figure 1B. ZEA reduced the viability of the cell in a dose-dependent and time-dependent manner. However, the cells treated with only with KFL alone did not differ in terms of viability from the control cells (Figure 1C), and in cells with KFL and ZEA, KFL protected cells from ZEA-induced cell death in a dose-dependent manner (Figure 1D). Overall, the above data showed that KFL potentially inhibits ZEA-induced hepatotoxicity.

### 2.2. ZEA Inhibits the Expression of Nrf2, HO-1 and NQO1 in HepG2 Cells

Nrf2 is an essential component in cells that is implicated in oxidative stress. During oxidative stress, Nrf2 can induce the expression of a variety of protective proteins (HO-1 and NQO1) to regulate oxidative stress [19]. We further assessed if ZEA would have an inhibitory role in Nrf2, HO-1 and NQO1 expression on the basis of Western blotting data, and the outcomes showed that ZEA significantly downregulated the expression of Nrf2, HO-1 and NQO1 in a dose-dependent manner, as demonstrated in Figure 2A.

### 2.3. Effects of KFL on Nrf2-Related Protein Expression in ZEA-Induced Hepatic Cells

Since KFL stimulates the Nrf2/HO-1/NQO1 pathway [20] (Appendix A), we assessed whether KFL activates the Nrf2/HO-1/NQO1 pathway in ZEA-supplemented cells. Based on Western blotting and qRT-PCR, it was found that the proteins were significantly (*p* < 0.05) upregulated in cells pre-treated with KFL, compared to cells treated with ZEA alone (Figure 2B and Figure 3A). These data suggest that KFL ameliorates ZEA-induced oxidative stress by upregulating Nrf2.

### 2.4. The Oxidative Stress Effect of KFL Is Attenuated by Nrf2 Knockdown

The signaling of Nrf2, which is crucially implicated in ROS-mediated hepatic fibrosis, intensifies oxidative stress and aggravates fibrotic pathology through downregulating Nrf2 expression [21]. To further confirm that KFL activates Nrf2 in ZEA-treated HepG2 cells, we performed knockdown studies and analysis Nrf2 and HO-1 expression by Western blotting. As Figure 3B demonstrates, Nrf2 expression was abolished in siRNA-transfected cells, even in cells pre-treated with KFL and ZEA. Based on this result, KFL may activate antioxidant systems via Nrf2 signaling.

### 2.5. KFL Could Potentially Up-Regulate Phospohrylation of PI3K/Akt Expression

Some of the key concerns are that inhibitors of pPI3K/pAkt increase liver and inflammation as well as the risk of hepatic cancer because hepatic elimination from Akt contributes to chronic liver damage, inflammation and hepatocellular carcinoma (HCC) [22]. Consequently, we further verified the influence of KFL in inhibiting ZEA-induced oxidative stress by PI3K/Akt signaling pathway through Western blotting. In the present study, we found that ZEA alone (40 µM) could downregulate pPI3K/pAkt activation, as shown in Figure 4A; however, after treatment with KFL and ZEA, the activation of the pPI3K/pAkt was significantly (*p* < 0.05) upregulated in a dose-dependent manner. Therefore, KFL could protect against ZEA-induced oxidative stress via the PI3K/Akt pathway.

### 2.6. KFL Activates PI3K/Akt Signalling to Regulate Nrf2 in ZEA-Induced Hepatic Cells

To testify if the up-regulation of pAkt is linked to with Nrf2 signals, stimulation of Akt protein was examined using Western blotting. Here, the selective inhibitor LY294002 was used to determine the crucial role of PI3K/Akt in the activation of Nrf2 signaling in KFL pre-treated cells with ZEA. As shown in Figure 4B, ZEA with KFL-treated cells has upregulated pAkt, Nrf2 and HO-1 compared to control cells; however, cells treated with LY294002, KFL and ZEA combined, had significantly downregulated levels of these proteins. Therefore, KFL may rely on the regulation of PI3K/Akt-Nrf2 signaling in HepG2 cells.

### 2.7. KFL Suppressed ZEA-Induced DNA Damage

Many studies have established that ZEA induces DNA damage and leads to apoptosis [23,24,25]. We examined if KFL is responsible for causing DNA damage induced by ZEA by Western blotting. We found that the expression of the DNA damage marker protein γ-H2A in ZEA-treated cells was significantly elevated (*p* < 0.05), while in KFL pre-treated cells also treated with ZEA, we found that γ-H2A protein expression is inhibited in a dose-dependent manner by KFL. These results indicate that KFL may inhibit DNA damage induced by ZEA in hepatic cells (Figure 5A). Additionally, augmented DNA fragmentation was noticed in KFL (50 µM) and/or ZEA-treated HepG2 cells (Figure 5B). ZEA-treated cells showed significantly elevated DNA fragmentation, on the other hand, in cells pre-treated with KFL, this effect was reversed to normal. These outcomes reveal that KFL can inhibit ZEA-induced DNA fragmentation in HepG2 cells.

### 2.8. KFL Inhibits Apoptosis in ZEA-Induced Hepatic Cells

Hepatic necrosis is a major indication of underlying liver failure. ZEA causes apparent apoptosis and necrosis at higher concentrations [26]. In the end, we also analyzed whether KFL could inhibit ZEA-induced apoptosis in hepatic cells. The results from Figure 5C show that ZEA markedly inhibited Bcl2 expression and subsequently elevated the levels of apoptosis marker proteins cleaved caspase-3 and cleaved PARP. The down-regulation of Bcl2 and up-regulation of cleaved protein levels were markedly reversed when cells were treated with KFL. These data suggested that KFL suppressed ZEA-induced apoptosis in HepG2 cells. The data from the TUNEL assay (Figure 5D) clearly show that KFL treatment can inhibit HepG2 cell apoptosis. Overall, the above-mentioned data demonstrate that KFL potentially inhibits ZEA-induced hepatotoxicity.

### 2.9. KFL Potentially Inhibits ZEA Induced toxic effects

When body weight and liver morphology there no statistically marked difference between the control and experimental groups (Appendix A). Appendix A showed animal behavioural studies, control animals have shown normal behaviour (Appendix A), ZEA alone treated animal has shown abnormal behaviour, agitated and fight with others (Appendix A), KFL with ZEA treated and KFL alone treated animals have shown normal behaviour and activity (Appendix A).

### 2.10. KFL Potentially Inhibits Pro-Inflammatory Cytokines

Comprehensive studies have shown that key genes mediate hepatic damage. Hepatocellular necrosis controlling genes and inflammation influence the degree of liver damage [27,28]. We also examined the anti-inflammatory influence of KFL on mice injected with ZEA using ELISA. The serum levels of IL-6, TNF-α and IL-β1 were noted to be significantly higher (*p* < 0.05) in mice treated only with ZEA, whereas IL-6, TNF-α and IL-1 β were inhibited by KFL. The levels of IL-6, TNF-α and IL-1β in KFL with ZEA-treated mice nearly reached the control levels (Figure 6B–D). KFL-treated mice showed no differences in comparison to the control mice. The obtained outcomes revealed that KFL has a potent cytoprotective effect against ZEA-induced toxicity.

### 2.11. KFL Inhibits Hepatic Marker Enzyme Activity

Aminotransferases (ALT and AST) are hepatocellular damage markers. As shown in Figure 6E–G, the ZEA-treated mice exhibited increased activities of AST, ALT and ALP compared to those of the control animals. However, ZEA with KFL-treated mice had markedly (*p* < 0.05) decreased activation of the above enzymes. Moreover, mice treated with KFL alone did not show any significant differences from control animals. This study demonstrated that KFL could mitigate ZEA-induced liver toxicity.

### 2.12. Effect of KFL on ROS Production

In various types of liver diseases, there is a marked involved of ROS-mediated oxidative stress. The endogenous antioxidants that subsequently fail to counteract all the ROS leading to cellular injury are excessively ROS-produced [29]. Hence, we have determined H_2_O_2_ production in liver control and experimental animals. The production of H_2_O_2_ was markedly (*p* < 0.05) augmented in ZEA-treated mice compared to control mice. In contrast, KFL with ZEA-treated mice showed downregulation of H_2_O_2_ production when compared to ZEA-treated mice (Figure 7A).

### 2.13. KFL Inhibited LPO

Figure 7B shows the level of LPO in liver control and treated mice. ZEA-treated mice exhibited elevated levels of LPO compared with control mice. While LPO was inhibited in KFL- and ZEA-treated mice, KFL-treated mice did not show any significant difference from the control mice. This result suggested that KFL potentially inhibits LPO in mice with ZEA-induced hepatotoxicity.

### 2.14. Effect of KFL on Antioxidant Enzymes

Mammalian cells, including hepatocytes, have developed a defense system against oxidative stress injury by enhancing the antioxidant enzymes. Levels of liver SOD, CAT and GSH were considerably low in the group in which ZEA was administered as compared to the control group (*p* < 0.05). These levels were markedly augmented in the KFL with ZEA-treated mice (*p* < 0.05 or *p* < 0.01) (Figure 7C–E). The above-mentioned results verified the suppressive role of KFL in ZEA-induced oxidative stress in liver.

### 2.15. Influence of KFLon Pathological Morphology of the Liver

As shown in Figure 8A, pathological alterations of the liver were inspected using H&E staining. Histopathological examination of liver sections in normal control group revealed normal hepatocyte distribution in linear hepatic cords with apparent nuclei, central vein and portal triad (panel a). In group treated with ZEA inflammatory cells were present with extensive inflammatory foci, and dispersed inflammation with severe necrosis of the tissue, interstitial edema, as well as loss of integrity. Inflammatory infiltrates were categorized into diffused and focal (panel b), we also found bi-nucleated hepatocytes, hemorragh in the sinusoids spaces, congestion of sinusoids and in the central vein, and hepatic cell death (panel c). For KFL with ZEA-treated groups, minimal necrosis, a significant decrease in sinusoidal congestion, cloudy swelling in the hepatic cells and areas of regeneration were noted (panel d). Mice in which KFL was injected constantly revealed inflammatory infiltrates and fibrotic lesions in negligible levels in the sections of liver which were similar to or lower compared to the control mice and retention of tissue integrity (panel c). These above data demonstrated that KFL enhanced hepato-protectivity effects in mice.

### 2.16. Effect of KFL on pPI3K, pAkt, Nrf2 and Bcl2 Expression in ZEA-Treated Liver Tissue

As seen from the Western blot analysis of pPI3K, pAkt, Nrf2 and Bcl2 in ZEA-treated liver tissue, these proteins were found to be downregulated, indicating that ZEA induced apoptosis by oxidative stress and that the activation of pPI3K, pAkt, Nrf2 and Bcl2 was effectively upregulated upon KFL treatment in the mouse liver (Figure 8B). Therefore, the results show that the effect of KFL on ZEA is mainly through regulating PI3K/Akt-mediated Nrf2 expression.

## 3. Discussion

The molecular mechanisms of mycotoxin-induced toxic effects have been identified, and it has been shown that oxidative stress and accumulation of free radicals are involved in mycotoxin toxicity [30]. Undoubtedly, the discrepancy between free radicals and antioxidant defense systems can lead to chemical impairments in DNA, lipids and proteins, as has been seen in mycotoxins exposure. In this study, we determined that KFL might be a novel factor influencing the pathological condition during ZEA-induced hepatic injury. In this investigation, we examined whether ZEA treatment had a significant impact on HepG2 cell viability. However, this oxidative stress was decreased with KFL treatment, indicating that ZEA increases oxidative stress and that KFL could significantly alleviate overproduction of oxidative stress signals. These data suggest that KFL could inhibit ZEA-induced oxidative stress cell death.

In the growth factor superfamily, PI3K is an essential signal transduction molecule. The phosphorylation of its serine and threonine residues activates Akt with the help of PI3K-dependent kinase (PDK). Numerous reports have shown that PI3K/Akt signaling can regulate ROS expression and cellular oxidative stress pathways [31,32,33]. It provides a survival signal that allows Keap-1 to release of Nrf2 and its succeeding translocation. It also regulates ROS-dependent activation of Nrf2 under certain stresses, such as oxidative stress. On the contrary, some researchers postulated that certain kinase pathways, including PI3K, can mediate the activation of Nrf2-ARE, independent of oxidative stress [34,35]. To examine whether KFL upregulates HO-1 through Nrf2 activation, protein was isolated, and Nrf2 protein expression was inspected through western blotting. The outcomes revealed that KFL supplementation noticeably augmented Nrf2 protein expression in the nucleus, suggesting that KFL may stimulate HO-1 expression by regulating Nrf2 signaling. We utilized Nrf2 knockdown cells to determine whether inhibiting Nrf2-regulated signaling ameliorated HO-1 expression induced by KFL. It was noted that inhibition of Nrf2 profoundly attenuated KFL-triggered HO-1 expression. Therefore, we suggested that KFL exerts antioxidant functions through upregulating HO-1 expression via Nrf2.

We hypothesized that PI3K/Akt regulates KFL in ZEA-treated cells. KFL supplementation markedly augmented the phosphorylation status of Akt protein, suggesting that heightening Akt protein phosphorylation may be implicated in KFL-stimulated HO-1 expression. To investigate the function of PI3K/Akt pathway, a specific PI3K/Akt inhibitor, LY29400, was used for HepG2 together with KFL with ZEA. The outcomes revealed that KFL inhibited PI3K/Akt and Nrf2 expression. Moreover, we assessed whether PI3K/Akt plays an essential role in decreasing ROS production via KFL. Fluorescence assessment demonstrated that blocking the PI3K/Akt with LY294002 ameliorated the ROS-decreasing ability of KFL in HepG2 cells in causing oxidative stress induced by ZEA, and this may be implicated in the downregulation of Nrf2 expression. Consequently, we suggest that the PI3K/Akt-signaling cascade plays an essential function in KFL-triggered HO-1 expression by mediating Nrf2 in ZEA-induced oxidative stress.

Mycotoxins, such as ZEA, could lead to immune system, oxidative and inflammatory injuries due to their physical and biological effects [36,37]. Close relations between oxidative stress, inflammatory reactions and the immune system (for example, TNF-α, IL-6 and IL-1β activation and increase) have been reported [37]. The current study reveals that KFL noticeably decreased TNF-α, IL-6 and IL-1β levels in animals in which ZEA was administered as compared to control animals. Thus, this decrease is caused by the inflammatory mediators of anti-inflammatory KFL mechanism.

Liver is a primary target organ for toxicity. ZEA myotoxicity is implicated in oxidative injury, which plays a crucial role in biochemical changes in the liver [38,39]. Loss of hepato-specific enzymes in blood serum has been considered an indicator of hepatic dysfunction and damage. Most importantly, ALT, AST and ALP are key markers in identifying hepatic injury [40]. In this study, we noticed a marked augmentation in ALT, AST and ALP levels in ZEA-injected mice compared to control mice, perhaps resulting from damage to the hepatocyte membrane. This may be due to ZEA-induced free radical damage in the lipid membrane of hepatocytes, as these enzymes in the cytosol are released into the bloodstream, indicating liver damage [41]. Nonetheless, pre-administration of KFL markedly decreased changes in these hepatic markers by ZEA via its membrane-stabilization properties against ROS-mediated oxidative hepatic injury. Our study suggests that KFL diminishes cellular oxidative stress/ROS levels and protects against liver injury.

Antioxidants can compete with other oxidizable substances at moderately low concentrations and considerably decrease oxidation. The physiological process of antioxidants is to provide a defense mechanism to the cellular components from injuries due to ROS and free radicals. Recent reports have stated that the accumulation of oxidative stress signals, free radicals and ROS plays a vital role in the development of numerous ailments, including cancer [42,43]. The protective effects of antioxidants, which are mostly of natural origin, have been noted to combat the toxic effects of numerous mycotoxins [30].

Studies in vivo and in vitro have demonstrated that ZEA increases MDA levels via induction of antioxidant mechanisms, such as decrease in activity of GSH and SOD and enhancing activities of GPx and CAT [37,44]. Latter enzymes are implicated in the intracellular antioxidant activity of H_2_O_2_; thus, the augmentation of GPx and CAT functions are relatable to the intracellular mechanism which diminishes ROS accumulation triggered by ZEA [37]. The ZEA-mediated decrease in the levels of antioxidants in hepatic tissue is well corroborated with our present results. This may be due to the high levels of ZEA-induced ROS generation and the formation of ZEA-glutathione conjugates, which causes severe depletion of glutathione and facilitates its subsequent reduction in the liver. Pre-administration of KFL has been shown to exert a protective mechanism by preventing the depletion of SOD, CAT and GSH, which leads to increased antioxidant levels. This may be due to the ability of KFL to protect against oxidative damage, inhibit membrane peroxidation and exhibit membrane stabilizing properties.

Hepatic injury stimulated via ZEA-caused oxidative stress causes apoptosis in cells. Caspase-3 and Bcl-2 are imperative proteins in the stimulation of the apoptotic pathway [45]. We inspected the mechanisms of action of KFL against ZEA-triggered cytotoxicity. The outcomes revealed that KFL partially reversed ZEA-mediated gene alterations in cleaved caspase-3, cleaved PARP and Bcl-2. Moreover, ZEA can cause HepG2 death by provoking cell apoptosis. Thus, it is feasible that KFL protects against ZEA damage by blocking the apoptotic pathway. KFL could also enhance the DNA repair pathway.

## 4. Experimental Section

### 4.1. Reagents

Agents used in this study are: 3-(4,5-dimethylthiazol-2-yl)-2,5-diphenyltetrazolium bromide or also known as MTT, LY294002 (L9908), Kaempferol (60010) and Zearalenone (Z2125) (Sigma-Aldrich (St. Louis, MO, USA). 2′7′-dichlorodihydrofluorescein diacetate (DCFH2-DA; CAS 4091-99-0) and deoxynucleotidyl transferase dUTP nick end labelling (TUNEL) staining kit (Roche, cat. no. 11684817910). Antibodies against pAkt (# 44-621G), Akt (# 44-609G), pPI3K (#PA5-104853), PI3K (# PA5-29220), BCL2 (# PA5-27094), PARP (PA5-16452), HO-1 (# PA5-77833), NQO1 (# PA5-82294), pNrf2 (# PA5-105664), β-actin (# PA5-78716) and Lamin B1 (# PA5-19468) were sourced from Invitrogen; Thermo Fisher Scientific, Inc. (Waltham, MA, USA). Anti-cleaved caspase-3 (ab32042) antibody was purchased from Abcam (Branford, CT, USA). Lipofectamine 2000 (11668027) and Nrf2 siRNA were procured from Thermo Fisher Scientific, Inc. (Waltham, MA, USA).

### 4.2. Cell Culture

HepG2 cells were obtained from the American Type Culture Collection (Virginia, USA), plated DMEM/F 12 medium containing 10% FBS (HyClone, Logan City, UT, USA) and cultured at 37 °C with 5% CO_2_.

### 4.3. Cell Culture Treatment

KFL and ZEA were suspended in serum-free medium. Cells were seeded onto plates and then with KFL, ZEA or a mixture was added over 24 h in triplicates in each group. Cells were added to a solution of 0.25% trypsin (w/v) and 0.52 mM EDTA (Cat. no: R001100, Thermo, Waltham, MA, USA).

### 4.4. Animals

#### 4.4.1. Ethics Statement

The care and maintenance of investigational animals were in compliance with the guidelines, ethical policies and procedures approved by the ethics committee of the King Faisal University (Approval no. 71507).

#### 4.4.2. Animal Experimental Design and Treatment

Twenty-four albino male mice weighing 25–30 gm were procured from Charles River Laboratories (Écully, France). The animals were maintained at a temperature of 22 ± 2 °C in a normal laboratory atmosphere with a 12/12 h light/dark cycle with access to food and water. Animals were randomly divided into 4 groups of 6 mice each. Each group comprised of six animals. The first group of animals was treated with saline (control), and the second group of animals was orally challenged with ZEA (40 mg/kg wt) from the second week to the end of the treatment process. The third group of animals was treated with KFL (50 mg/kg wt) alone in the first week, and from the second week, each animal was treated for 14 days by oral gavage (without anesthesia) with ZEA (40 mg/kg wt) and with KFL (50 mg/kg wt) 4 h later. The fourth group served as the drug control, and animals were treated with KFL from the second week to the end of the experiment.

Animals were sedated by administering ether and were sacrificed at the experiment endpoint. Blood was collected via cardiac puncture, and serum was separated and stored at −80 °C for pro-inflammatory cytokine and biochemical analyses. Livers were excised and weighed accurately with the help of a sensitive balance (Nimbus, MK, UK). The livers were homogenized in 0.1 M Tris–HCl buffer (pH 7.4) for subsequent assessment of biochemical parameters.

### 4.5. Viability Assay

MTT assay was used to assess the viability of the cells. These cells were then maintained in wells of a 12-well plate, at 4 × 10^5^ cells/well, and exposed to various treatments. This was followed by further incubating the cells for 2 h of PBS containing 400 μL of MTT (0.5 mg/mL). After removing medium, the developed formazan product was liquefied in DMSO (400 μL). Finally, a microplate reader (Winoosky, VT, USA) was used to determine the absorbance of each well at 570 nm. Untreated cells were used as a control for cell viability (set at 100%).

### 4.6. Preparation of Cytoplasmic and Nuclear Extracts

The cell pellets were resuspended in Buffer I for 5 min to prepare the cytoplasmic extracts. Buffer I consisted of 25 mM HEPES pH 7.9, 5 mM KCl, 0.5 mM MgCl_2_, and 1 mM dithiothreitol (DTT). This suspension was then mixed with the same amount of Buffer II consisting of 25 mM HEPES pH 7.9, 5 mM KCl, 0.5 mM MgCl_2_, and 1 mM DTT. Additionally, protease and phosphatase inhibitors supplemented with 0.4% (v/v) NP40 were mixed with the suspension. The obtained suspension samples were incubated at 4 °C for 15 min with rotation. Then lysates were centrifuged at 2500 rpm at 4 °C for 5 min in a microfuge. After that, the supernatants were transferred to new Eppendorf tubes. Buffer II was used to clean the pellets once, and the supernatant was added to the cytoplasmic protein tube. To remove residual nuclei, again, the lysates were centrifuged again at 4 °C at 10,000× *g* for 5 min, and they were transferred to new Eppendorf tubes.

The pellets obtained from cytoplasmic extraction were subjected to incubation with Buffer III for nuclear extraction. In addition to protease and phosphatase inhibitors, Buffer III comprised of HEPES 25 mM, pH of 7.9, NaCl—400 mM, 10 percent of sucrose or dextrose, 0.05% NP-40, and 1 mM DTT. The lysates were rotated at 4 °C for 1 h and then centrifugation was carried out at 4 °C at 1000 rpm for 10 min. After this step, the collected supernatants contained nuclear proteins.

### 4.7. siRNA Transfection

Transfection with Nrf2 siRNA was performed with a 5′–3′ sequence targeting human Nrf2 siRNA. HepG2 cells were loaded at 1.5 × 10^5^ cells per well into 6-well plates, and transfection was conducted using Lipofectamine 2000, based on the instructions in the manufacturer’s protocol. Shortly, an appropriate amount of Nrf2 siRNA and 5 μL Lipofectamine 2000 in 250 μL serum-free DMEM/12 medium was prepared in individual RNase-free tubes. After 5 min incubation, siRNA and Lipofectamine were combined and incubated for 20 min and supplemented to each well. After incubation with 100 pM siRNA per well for 24 h, KFL and/or ZEA was added to the cells for 24 h for protein analysis.

### 4.8. Western Blotting

After treatment, cells were harvested and then washed once with cold PBS, after which cytoplasmic, nuclear and total extracts were prepared in the aforementioned manner. In each sample, to detect the protein status, a Bio-Rad protein assay was used, and bovine serum albumin (BSA) was used as the reference standard. SDS-PAGE (8–15%) was used to resolve equal amounts of protein (50 μg), and the proteins were transferred to nitrocellulose membranes overnight. Five-percent skimmed milk was used for blocking the membranes at 37 °C for 30 min, after which the membranes were readily incubated with the indicated primary antibodies for 2 h. After this, a horseradish peroxidase-conjugated goat anti-mouse or anti-rabbit secondary antibody was incubated with the nitrocellulose membranes for 1 h, and an enhanced chemiluminescence substrate was used to develop the membranes (Pierce Biotechnology, Rockford, IL, USA). A LI-COR chemiluminescence imaging system (3600-00-C-Digit Blot Scanner) was used to examine the samples. Image Studio Lite software (LI-COR Biosciences, Lincoln, NE, USA) was used to generate the graphs of the densitometric band intensities with normalization to the intensity of the untreated control band, which was set to 1.

### 4.9. DNA Fragmentation Assay

One of the hallmarks of apoptotic cell death is the fragmentation of nuclear DNA into nucleosomal units. It occurs in response to various apoptotic stimuli in a wide variety of cell types. Here we determine the DNA fragmentation in HepG2 cells upon KFL and/or ZEA treatment, the Cell Death Detection ELISA PLUS kit (Roche Applied Science, Branford, CT, USA) was used as per to the recommendation of the manufacturer and as mentioned previously [46,47].

### 4.10. TUNEL Assay

HepG2 cells at the logarithmic growth stage were loaded in a six-well plate and supplemented with ZEA or KFL. After that, the medium was removed, and the cells were cleaned with PBS and processed with 4% paraformaldehyde for 20 min and the paraformaldehyde was removed, the cells re-washed with PBS. Afterwards, the cells were incubated with TUNEL reagent (11684817910, Roche, Mannheim, Germany). Following to the PBS wash, cells were counterstained with 0.1 μg/mL DAPI for 5 min. Finally, cells were examined under a fluorescence microscope. All morphometric studies were executed three times. TUNEL-positive cells were detected as brilliant green, and the cell nuclei were detected through UV light microscopy at 454 nm. Images were obtained through microscopy (200× magnification), and a Leica D6000 fluorescence microscope was used for measurement (Leica, Wetzlar, Germany).

### 4.11. Real-Time Polymerase Chain Reaction (RT-PCR)

The KFL-injected cells were cleaned with PBS and tRNA was isolated from HepG2 cells using TRIzol reagent (Invitrogen, Carlsbad, CA, USA). We then converted RNA to cDNA using a PrimeScript RT reagent kit as per the recommended guidelines of the manufacturer (Takara Bio, Shiga, Japan). Real-time qPCR was performed using SYBR Green system (Applied Biosystems, Foster City, CA, USA) and a ViiA-7 Applied Biosystem (Carlsbad, CA, USA). The mRNA expression in all genes was standardized to the expression of β-actin housekeeping gene. The primer sequences were as follows: Nrf2: sense, 5′-CATCCAGTCAGAAACCAGTGG-3′ and antisense, 5′-GCAGTCATCAAAGTACAAAGCAT-3′; HO-1: sense, 5′-CTTCTTCACCTTCCCCAACA-3′ and antisense, 5′-ATTGCCTGGATGTGCTTTTC-3′; NQO1: sense, 5′- GGGATCCACGGGGACATGAATG-3′ and antisense, 5′-ATTTGAATTCGGGCGTCTGCTG-3′; and β-actin: sense, 5′-GGAAATCGTGCGTGACATTA-3′ and antisense, 5′-GGAGCAATGATCTTGATCTTC-3′. The mRNA expression status (fold change) was determined between groups by 2-ΔΔCt value compared with the non-treat (NT) samples was determined.

### 4.12. Analysis of Pro-Inflammatory Cytokines

The cytokine status of serum was inspected by implementing a sandwich ELISA (enzyme-linked immunosorbent assay) technique using commercial kits (Thermo Fisher Scientific Co., Waltham, MA, USA). TNF-α, IL-6 and IL-β1 in the serum were investigated via commercial cytokine ELISA kits (cat# KMC0061 for IL-6; cat# KMC3011 for TNF-α; cat # BMS6002 IL-β1) obtained from Invitrogen (USA). Cytokine status was inspected in regards with the recombinant biotinylated murine TNF-α and IL-6 standards supplied with the kits. These kits were manufactured by Endogen-Pierce and distributed by Fisher Thermo Scientific Co., and all controls and standards mentioned in the manufacturer’s kit were implemented.

### 4.13. Biochemical Analysis

The liver was excised and homogenized (10% w/v) with chilled 0.1 M sodium phosphate buffer (pH 7.4). Then, the homogenate was centrifuged twice at 11200 rpm for 20 min at 4 °C to obtain an enzyme fraction. The supernatant was used for biochemical studies. The aspartate aminotransferase [48], alanine aminotransferase (ALT) and alkaline phosphatase [49] activities were inspected using the manufacturer’s protocols (Spectrum Diagnostics, Obour, Egypt). GSH, SOD and CAT assay kits (Cayman Chemical, Ann Arbor, MI, USA) were utilized for analysis.

#### 4.13.1. Measurement of H_2_O_2_

The amounts of H_2_O_2_ were measured by H_2_O_2_ assay kit (cat. no. AB102500). In the presence of Horse Radish Peroxidase (HRP), the OxiRed Probe reacts with H_2_O_2_ to produce product with color. In 96-well plates, normal diluents (100 μL), lysates collected from each sample (100 μL) and 100 μL of reaction mixture (50 μL working enzyme solution and 50 μL probe) were added; the plates were then incubated for 1 h at room temperature on a plate shaker. At 590 nm, the optical density was read.

#### 4.13.2. Measurement of LPO

The levels of LPO were detected by LPO kit ((cat. no. ab133085). The lipid hydroperoxide (LPO) content was determined in the liver prepared from ZEA-treated mice pretreated with or without KFL using the commercial LPO assay package. Calculating malondialdehyde (MDA) and 4-hydroxy nonenal (4-HNE), degradation products of polyunsaturated fatty acids (PUFAs) hydroperoxides, lipid peroxidation is quantified at 500 nm, the optical density was read.

#### 4.13.3. Measurement of GSH

Total intracellular glutathione (GSH+GSSG) using a glutathione assay kit based on Ellman’s reagent, according to the manufacturer’s protocol (Cat# 703002, Cayman Chemical, Ann Arbor, MI, USA). GSH concentration of samples determined by end-point method (read the plate at 450–414 nm after 25 min).

### 4.14. Histopathology Examination

The harvested livers of mice were fixed in formalin, dried with 100%, 95% and 75% alcohol, and implanted in paraffin wax. Tissue slices (0.2 μm thick) were washed in xylene to eradicate paraffin and spotted with haematoxylin and eosin (H&E). Finally, the slices were cleaned with water to eliminate excess H&E before they were stained with Masson’s trichrome stain to assess the morphological and fibrotic alterations in the liver, where blue staining indicated collagen thickening.

### 4.15. Statistical Analysis

Statistical analyses were performed using GraphPad Prism software version 6.0 (GraphPad Software Inc., San Diego, CA, USA), and for comparison of three groups, one-way ANOVA was used. Data are represented as the mean ± SD, and *p* < 0.05 was regarded as significant.

## 5. Conclusions

Our results showed that using kaempferol as an additive could effectively remove ZEA contamination in feed and protect against the toxic damage of ZEA both in vivo as well as in vitro. The findings of the current research revealed KFL’s therapeutic and prophylactic efficacy against ZEA-promoted oxidative hepatic damage via its strong antioxidant properties. KFL ameliorates ZEA-induced alterations in the liver via Nrf2 activation by PI3K/Akt-mediated pathway. Moreover, Nrf2 has emerged as a target factor for a wide variety of genes which are beneficial in detoxification and elimination of ZEA-induced liver oxidative damage. Further in-depth studies may establish bioactive KFL as a potential contender for the treatment of ZEA-induced oxidative stress—with hepatic complications in the future.

## Figures and Tables

**Figure 1 ijms-22-00217-f001:**
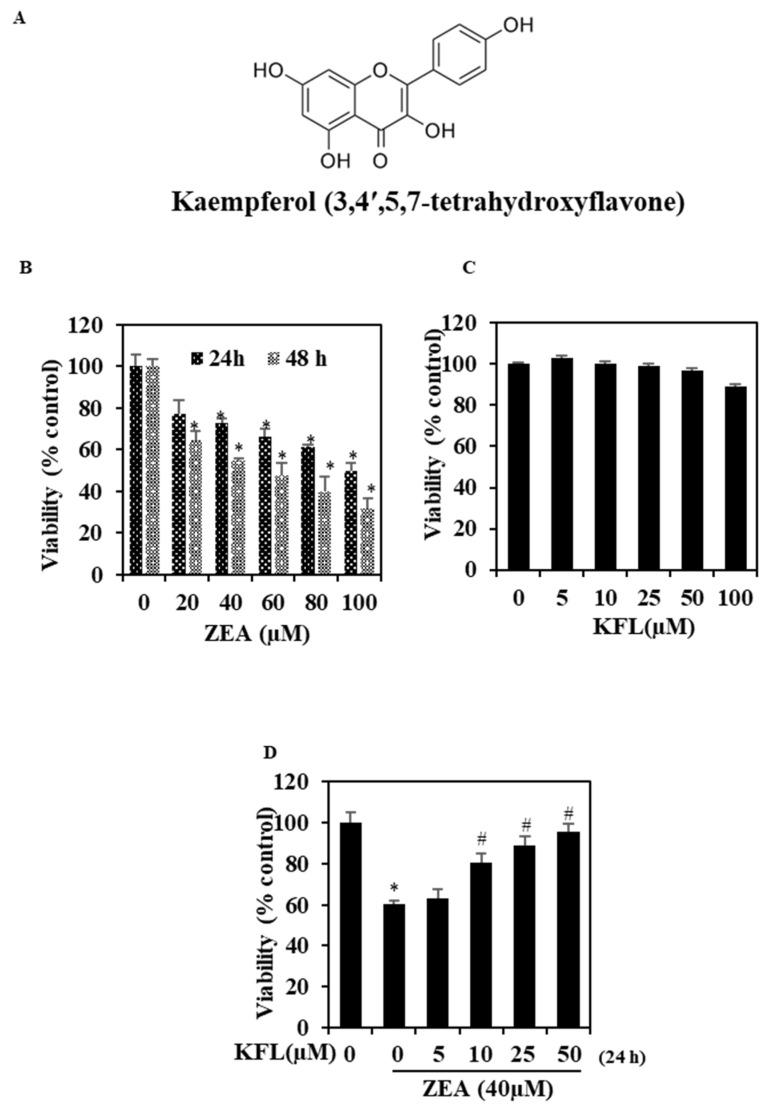
Impact of zearalenone (ZEA) and kaempferol (KFL) on viability. (**A**) Chemical structure of kaempferol. (**B**) HepG2 cells were added with the indicated dosages of ZEA for 24 h and 48 h, and then viability was determined through MTT assay. (**C**) Cytotoxic effect of KFL on HepG2 cells. Cells were exposed to different concentrations of KFL for 24 h. (**D**) KFL protects the cytotoxic effect of ZEA (24 h) as analyzed by MTT assay. Data are represented as the mean ± SD of triplicate values (*n* = 3), and * *p* < 0.05 represents significant difference in comparison to the control group. # *p* < 0.05 represents significant variations compared the ZEA alone and KFL with ZEA treatment groups.

**Figure 2 ijms-22-00217-f002:**
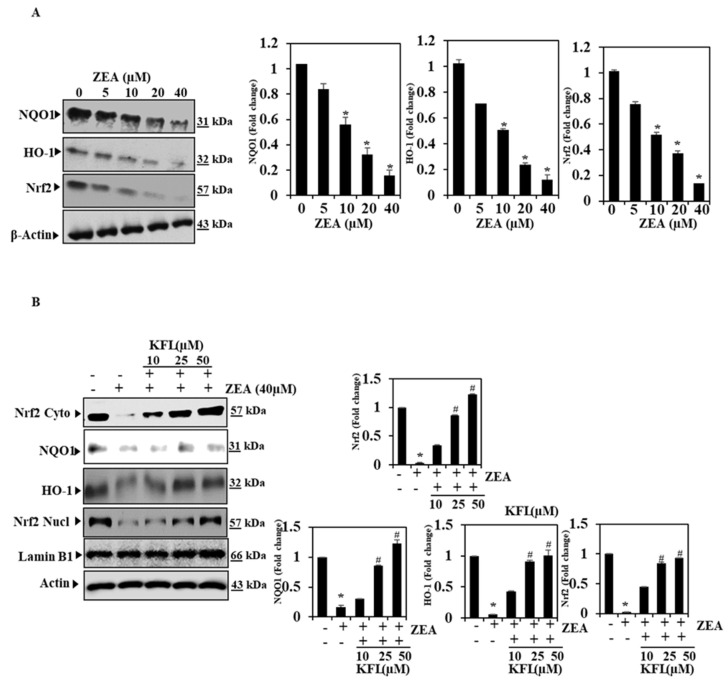
KFL induces nuclear factor-erythroid 2-related factor 2-dependent haem oxygenase-1 (HO-1) expression in ZEA-treated HepG2 cells. (**A**) Cells were incubated with various concentrations of ZEA for 24 h, and equal amounts of whole-cell lysate were exposed to SDS-PAGE. Membranes probed with anti-NQO1, anti-HO-1 and anti-Nrf2 antibodies. (**B**) Cells were treated with KFL (50 μM) and/or ZEA (40 μM) for 24 h, and the Western blotting results show the effects of KFL on the protein levels of HO-1, NQO-1, and Nrf2 in the cytosolic and nuclear fractions. The relative ratios of expression in the Western blotting results are demonstrated below each of the results as values relative to actin expression. Data are represented as the mean ± SD of triplicate values (*n* = 3), and * *p* < 0.05 represents significant variations compared with the control. # *p* < 0.05 represents remarkable variations compared the ZEA alone and KFL with ZEA treatment groups.

**Figure 3 ijms-22-00217-f003:**
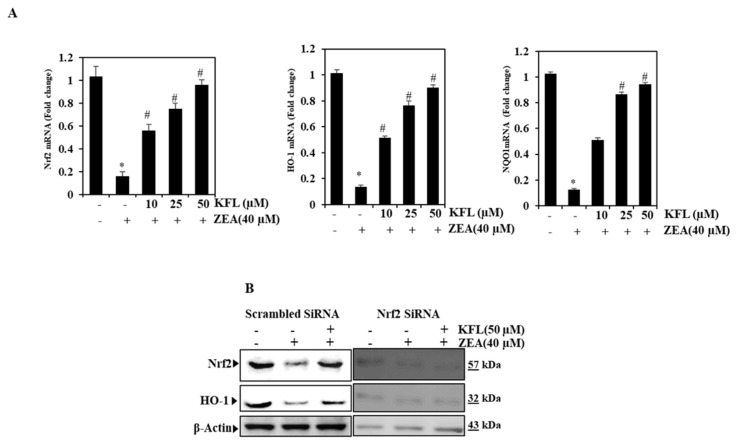
KFL up-regulates the antioxidant markers Nrf2, NQO-1 and HO-1 in HepG2 cells. (**A**) Nrf2, NQO-1 and HO-1 expression analyzed by RT-PCR. (**B**) The effects of inhibiting Nrf2 by siRNA transfection were studied by protein analysis. Cells were transiently transfected with Nrf2 siRNA at 100 pM per well in 6-well plates for 24 h, followed by treatment with ZEA and/or KFL for 24 h. SDS-PAGE was used to detect the expression of Nrf2 and HO-1 proteins. Data are represented as the mean ± SD of triplicate values (*n* = 3), and * *p* < 0.05 denotes remarkable variations as compared to the control. # *p* < 0.05 denotes remarkable variations compared the ZEA alone and KFL with ZEA treatment groups.

**Figure 4 ijms-22-00217-f004:**
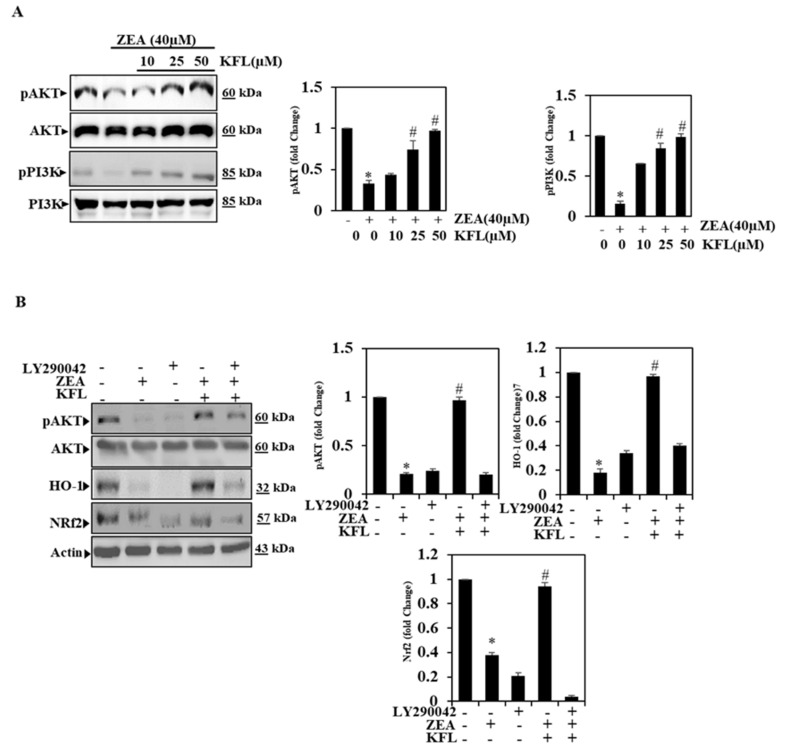
KFL triggers the PI3K/Akt pathway in ZEA-induced HepG2 cells. (**A**) Cells were treated with KFL (10, 25 and 50 µM) followed by ZEA (40 µM) for 24 h. After the treatment, whole-cell lysates were exposed to Western blotting with anti-pPI3K and anti-pAkt antibodies. Total PI3K and Akt levels were measured as loading controls. (**B**) Cells were pre-treated with a PI3K/Akt inhibitor (LY294002, 30 μM) for 2 h, followed by KFL (50 μM) and/or ZEA (40 μM) for 24 h. Western blot was performed to detect the pAkt, HO-1 and Nrf2 levels by anti-pAkt, anti-HO-1 and anti-Nrf2 abs. Data are represented as the mean ± SD of triplicate values (*n* = 3), and * *p* < 0.05 represents noteworthy discrepancies compared with the control. # *p* < 0.05 represents significant variations compared with the ZEA alone and KFL with ZEA treatment groups.

**Figure 5 ijms-22-00217-f005:**
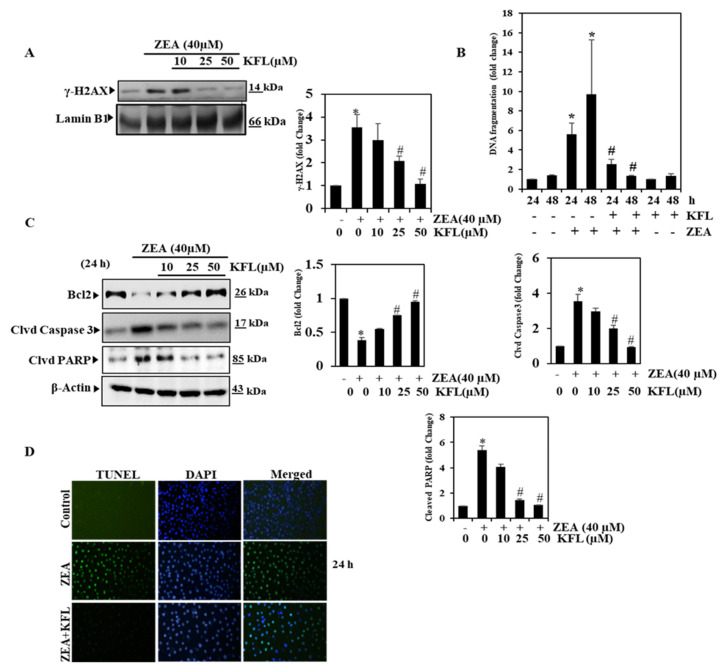
KFL repressed DNA damage and apoptosis induced by ZEA in HepG2 cells (**A**) γ-H2AX nuclear protein expression analyzed and γ-H2AX normalized to Lamin B1. (**B**) DNA fragmentation (**C**) HepG2 cells were treated with the KFL and/or ZEA for 24 h. Expression of BCL2, cleaved caspase-3 (clvd Caspase3) and cleaved PARP (clvd PARP) protein was assessed by Western blotting. (**D**) KFL protected against ZEA-induced apoptosis (24 h), as indicated by TUNEL assay. Data are represented as the mean ± SD of triplicate values (*n* = 3), and * *p* < 0.05 represents significant variations compared with the control. # *p* < 0.05 represents significant variations as compared to ZEA alone and KFL with ZEA treatment groups.

**Figure 6 ijms-22-00217-f006:**
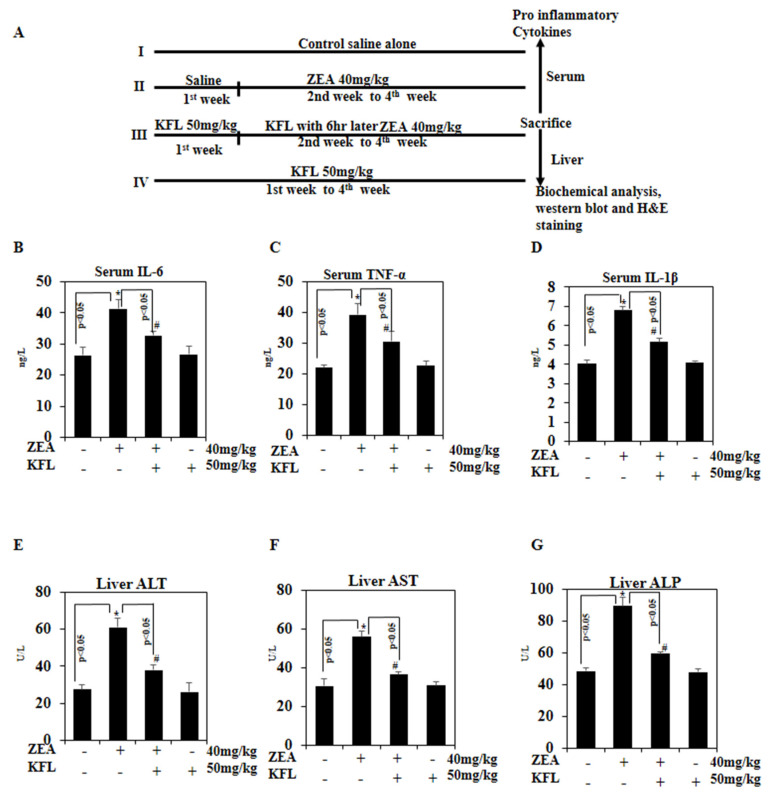
Effect of KFL on pro-inflammatory cytokines in ZEA-treated mice. (**A**) Schematic of the animal experimental method. (**B**–**D**) ELISA was used to quantity the levels of the pro-inflammatory cytokines (**B**) IL-6, (**C**) TNF-α and (**D**) IL-1β in mouse serum. (**E**–**G**) The level of liver markers was analyzed by ELISA: (**E**) ALT, (**F**) AST and (**G**) ALP. Data are expressed as the mean values ± SD of independent experiments (*n* = 6); * *p* < 0.05 represents significant variations in the ZEA alone group compared with the control group. # *p* < 0.05 represents significant variations compared the ZEA alone and KFL with ZEA treatment groups.

**Figure 7 ijms-22-00217-f007:**
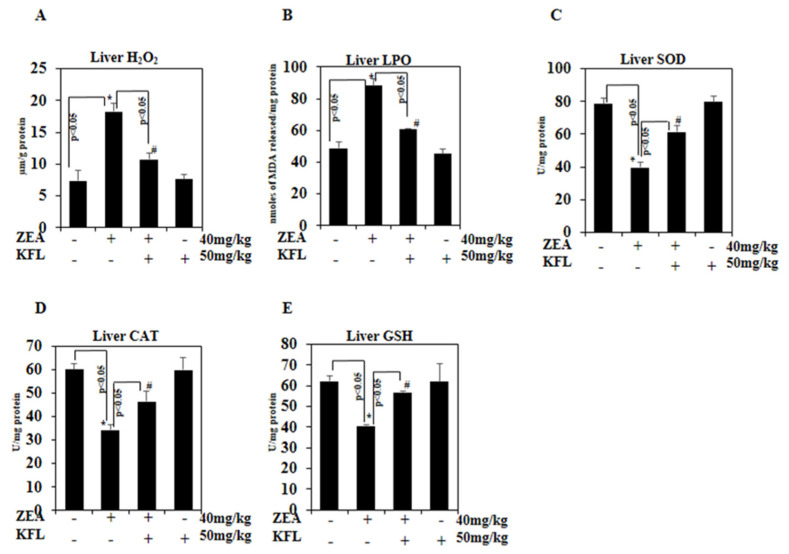
Effect of KFL on ZEA-induced ROS accumulation and antioxidant systems in mouse livers. (**A**) H_2_O_2_, (**B**) LPO, (**C**) SOD, (**D**) CAT and (**E**) GSH. ELISA kits were used based on the manufacturer’s directions (units: H_2_O_2_, µm/g protein; LPO, nM of MDA released/mg protein; SOD, CAT and GSH, U/mg protein). Data are expressed as the mean values ± SD of independent experiments (*n* = 6). * Significant in the ZEA alone group compared with the control group (*p* < 0.05). # *p* < 0.05 represents significant variations compared the ZEA alone and KFL with ZEA treatment groups.

**Figure 8 ijms-22-00217-f008:**
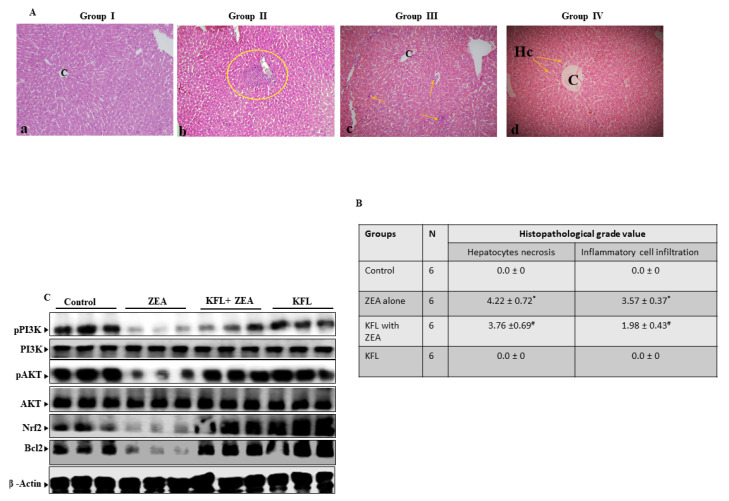
Effect of KFL on histopathological changes in control and experimental animals. (**A**) Microscopic pictures of mouse liver tissue (H&E, 400×). Representative images of H&E staining of liver tissue from mice after treatment with ZEA alone or with both ZEA and KFL: panel (**a**)—control, hepatocyte cords (Hc), clearly visible nuclei of hepatocytes, central vein (**C**) and normal hepatocytes visible panel (**b**,**c**)—ZEA alone, (**B**)—Circle signifies infiltrated inflammatory cells, panel (**c**)—KFL with ZEA, lesser sinusoidal congestion and some infiltrated inflammatory cells and panel (**d**)—KFL alone, marked arrow showed the normal hepatocytes and decrease of sinusoidal congestion, reduction in hepatocytes swelling exhibit recovery and retention of tissue integrity. (**B**) The histopathological grade values of the hepatocyte necrosis and inflammatory cell infiltration into the liver at the end of the experiment were measured (n = 6). (**C**) Inflections in pPI3K, pAkt, Nrf2 and Bcl2 proteins in albino mice. β-Actin was used as the internal control. Data are articulated as the mean values ± SD of independent experiments (n = 6). * Significant in the ZEA alone group compared with the control group (*p* < 0.05). # *p* < 0.05 represents significant variations compared to the ZEA alone and KFL with ZEA treatment groups.

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
