# Peer review of "Kaempferol Inhibits Zearalenone-Induced Oxidative Stress and Apoptosis via the PI3K/Akt-Mediated Nrf2 Signaling Pathway: In Vitro and In Vivo Studies"

_ijms, 2020, doi:10.3390/ijms22010217_

Round 1

Reviewer 1 Report

The purpose of the publication is to demonstrate that an antioxidant compound,  kaempferol, is able to counteract the oxidative stress induced by the mycotoxin zearalenone. In that view the authors used hepatic HepG2 cells and eventually investigate that matter in vivo in a mouse model. More specifically, they intend to demonstrate that kaempferol acts through the PI3/AKT-mediated Nrf2 pathway.

Within the whole manuscript, the authors used “µmol” instead of µM to indicate concentrations of compounds. This must be corrected all along the text.

2- Experimental section

The experimental section does not describe how were cultured the HepG2 cells.

Part 2.6 (intracellular ROS assay) describes the assessment of ROS content using H2DCF-DA and fluorescence microscopy. However, transformation of H2DCF-DA to fluorescent DCF mainly reveals H2O2 content and not every type of ROS. The authors indicate that they measure intracellular ROS but they do not explain how they obtain a quantification of the fluorescence from the images.

Part 2.10 should sum up the basic principles of the DNA fragmentation assay.

Part 2.14 must clearly explain the methods to measure H2O2, LPO and GSH.

3- Results

3-1 

The time of exposure with ZEA 40µM and KFL (Figure 1 –D) must be specified. It seems to be 24hr since the time of exposure with KFL alone is 24hr. The authors must precise that point in the legend.

Additionally, Fig 1-D must show a data relative to the exposure to ZEA without KFL.

Indeed, Fig. 1-D is troubling: it is not clear whether the data “0” represents exposure with or without ZEA. The authors must redesign that figure to make it understandable.

Fig 1-D must show results with KFL alone. Time of exposure must be indicated.

3-2 and 3-3 

The statistical analyses of Fig-2 and 3-A specify the significant difference between the control and only the highest concentration of ZEA or KFL.  However, to demonstrate that a compound acts in a dose dependent manner, the statistical analysis must show the difference of effect between each dose.

Fig 2-B must include a lane regarding a treatment with KFL alone.

In 3-3 the authors assume that KFL stimulate the Nrf2/HO-1/NQO1 pathway without giving a reference. It must be added.

3-4

Since scrambled siRNA is a control v.s. Nrf2 siRNA, it is wrong to say that the control cells “exhibit upregulation of Nrf2 and HO-1”. This scrambled siRNA treated cells should be compared with not treated cells to assume that. Moreover it seems that the presence of KFL decrease the level Nrf2 and HO-1 (fig 3-B) in the scrambled siRNA treated cells. Can the authors comment that results which is contrary to the preceding results ?

3-5

In Fig 3-C, an image of cells treated with NAC and ZEA is lacking to compare the effect of KFL with NAC.

Quantification of the fluorescence intensity in the cells should help to analyze the results.

3-6 and 3-7

Western-blots show that ZEA induced a decrease of the phosphorylated forms of AKT and PI3K (and not the no-phosphorylated forms) and that KFL can reverse that effect in a dose dependent manner.

However, the authors conclude that “ZEA could potentially down-regulate PI3/AKT expression” while it is not supported by their results. The authors must reconsider their comments to be in accordance with the experimental data. A control with cells treated with KFL alone is still missing in Fig 4.

Statistical analysis of Fig 4 is still incomplete.

3.9

Time of exposure to manage the experiment relative to apoptosis must be indicated.

Fig5-A : a control with the effect of KFL alone is still lacking

Fig5-B: the results should be normalized with the number of cells to be able to assess the effect of ZEA on DNA fragmentation. How did the authors process to take that point in consideration?

Results regarding PARP (Fig 5-C) are not consistent: lanes of western blot and histogram show opposite results!!

Size of the f PARP fragment must be indicated on the WB.

3-10

The authors indicate that “IL-6, TNF-a and IL-b1 were suppressed by KFL” while the results show that KFL inhibited the increase of this cytokines induced by ZEA.

Author Response

Reviewer 1

The purpose of the publication is to demonstrate that an antioxidant compound, Kaempferol, is able to counteract the oxidative stress induced by the mycotoxin Zearalenone. In that view the authors used hepatic HepG2 cells and eventually investigate that matter in vivo in a mouse model. More specifically, they intend to demonstrate that kaempferol acts through the PI3/AKT-mediated Nrf2 pathway.

 Thank you for giving us the opportunity to submit a revised draft of our manuscript entitled “Kaempferol inhibits zearalenone-induced oxidative stress and apoptosis via the PI3/AKT-mediated Nrf2 Signaling pathway: in vitro and in vivo studies” [Manuscript ID: IJMS 1012156] to International Journal of Molecular Science.

We appreciate the time and effort that you and the reviewers have dedicated to providing valuable feedback on our manuscript. We are grateful to the reviewers for their insightful comments, which have improved the paper.

We have incorporated changes that reflect all the suggestions provided by the reviewers. All changes are highlighted in green in the revised manuscript. Please see below for our point-by point response to the reviewers’ comments. If any responses are unclear or you wish additional changes, please do not hesitate to let us know.

Within the whole manuscript, the authors used “µmol” instead of µM to indicate concentrations of compounds. This must be corrected all along the text.

Thank you for your suggestions: Our revised manuscript is formatted as you suggested

2- Experimental section

The experimental section does not describe how were cultured the HepG2 cells.

Thank you for your comments: Our revised manuscript is formatted as you suggested

Part 2.6 (intracellular ROS assay) describes the assessment of ROS content using H2DCF-DA and fluorescence microscopy. However, transformation of H2DCF-DA to fluorescent DCF mainly reveals H2O2 content and not every type of ROS. The authors indicate that they measure intracellular ROS but they do not explain how they obtain a quantification of the fluorescence from the images.

Thank you for your valuable comments: in our revised manuscript, we have described how to measure the intracellular ROS and how to obtain quantification of the fluorescence from the images.

Part 2.10 should sum up the basic principles of the DNA fragmentation assay.

Thank you for your comments: The above-mentioned data has been added to our revised manuscript (line no: 176-178)

Part 2.14 must clearly explain the methods to measure H2O2, LPO and GSH.

Thank you for your comments: The above-mentioned data has been added to our revised manuscript , as you suggested. (line no: 223-242)

3- Results

3-1 

The time of exposure with ZEA 40µM and KFL (Figure 1 –D) must be specified. It seems to be 24hr since the time of exposure with KFL alone is 24hr. The authors must precise that point in the legend.

Thank you for your  comments, in our revised manuscript we have precised that point in the figure legends as suggested. (line no:271)

Additionally, Fig 1-D must show a data relative to the exposure to ZEA without KFL.

Thank you for your comments. In Fig 1D, We have included ZEA (source: Fig 1B) alone as per your suggestion.

Indeed, Fig. 1-D is troubling: it is not clear whether the data “0” represents exposure with or without ZEA. The authors must redesign that figure to make it understandable.

Fig. 1-D was redesigned according to your suggestions. Thank you

Fig 1-D must show results with KFL alone. Time of exposure must be indicated.

 In our revised manuscript, time exposure was indicated in figure 1-D as suggested.

3-2 and 3-3 

The statistical analyses of Fig-2 and 3-A specify the significant difference between the control and only the highest concentration of ZEA or KFL.  However, to demonstrate that a compound acts in a dose dependent manner, the statistical analysis must show the difference of effect between each dose.

The statistical test utilized has been described in each figure legend as suggested by reviewer.

Fig 2-B must include a lane regarding a treatment with KFL alone.

Thank you for your valuable comments. We have done already KFL alone dose dependent expression of Nrf2, HO-1 and NQO-1 but we didn’t include it in the main manuscript results (Please find it as supplementary file in fig. 2).

In 3-3 the authors assume that KFL stimulate the Nrf2/HO-1/NQO1 pathway without giving a reference. It must be added.

 The reference was added as you suggested (line no: 284)

3-4

Since scrambled siRNA is a control v.s. Nrf2 siRNA, it is wrong to say that the control cells “exhibit upregulation of Nrf2 and HO-1”. This scrambled siRNA treated cells should be compared with not treated cells to assume that. Moreover it seems that the presence of KFL decrease the level Nrf2 and HO-1 (fig 3-B) in the scrambled siRNA treated cells. Can the authors comment that results which is contrary to the preceding results ?

Thank you for your comments. The figure 3B was redesigned, because there was a mistake in (+) and (-) positions.

3-5

In Fig 3-C, an image of cells treated with NAC and ZEA is lacking to compare the effect of KFL with NAC. Quantification of the fluorescence intensity in the cells should help to analyze the results.

 In our revised manuscript , fluorescence intensity figure was added as suggested

3-6 and 3-7

Western-blots show that ZEA induced a decrease of the phosphorylated forms of AKT and PI3K (and not the no-phosphorylated forms) and that KFL can reverse that effect in a dose dependent manner.

Thank you for your comments. As suggested, in our revised manuscript, we have modified the above mentioned data.  (line no: 332-334)

However, the authors conclude that “ZEA could potentially down-regulate PI3/AKT expression” while it is not supported by their results. The authors must reconsider their comments to be in accordance with the experimental data. A control with cells treated with KFL alone is still missing in Fig 4.

We made the correction as per your suggestion.

Statistical analysis of Fig 4 is still incomplete.

 The statistical test utilized has been described in each figure legend as suggested by reviewer

3.9

Time of exposure to manage the experiment relative to apoptosis must be indicated.

In our revised manuscript, we have indicated time exposure in the legends. (line no:401)

Fig 5-A : a control with the effect of KFL alone is still lacking

We have included KFL alone dose dependent manner in supplementary in fig 2B

Fig 5-B: the results should be normalized with the number of cells to be able to assess the effect of ZEA on DNA fragmentation. How did the authors process to take that point in consideration?

Thank you for your valuable comments, we have found DNA fragmentation significant at elevated levels in 24h treatment with ZEA when compared to control (24h).  In 48h, control cells also found little amount fragmentation due to over cell growth.  

Results regarding PARP (Fig 5-C) are not consistent: lanes of western blot and histogram show opposite results!!

We are sorry for that incorrected figure, in our revised manuscript we have included the corrected histogram.

Size of the f PARP fragment must be indicated on the WB.

Our revised manuscript size of the cleaved PARP was included as suggested

 3-10

The authors indicate that “IL-6, TNF-a and IL-b1 were suppressed by KFL” while the results show that KFL inhibited the increase of this cytokines induced by ZEA.

Thank your comments. We have corrected the above-mentioned details in our revised manuscript. (line no:409)

Thank you for all your valuable comments and questions, which allowed us to improve the quality of the manuscript.

Reviewer 2 Report

The present study is well designed and very well prepared. The results are convincing as well. I clearly recommend this work for publication in IJMS. However, prior to publish this manuscript required minor revision must be done.

They are…

  1. There are number of typos were found throughout the manuscript. I advise the author should provide extra attention for those typos (Please find the attached file which I have indicated those typos.
  2. The Figures quality must be improved.

Author Response

Reviewer 2

The present study is well designed and very well prepared. The results are convincing as well. I clearly recommend this work for publication in IJMS. However, prior to publish this manuscript required minor revision must be done.

Thank you for giving us the opportunity to submit a revised draft of our manuscript entitled “Kaempferol inhibits Zearalenone-induced oxidative stress and apoptosis via the PI3/AKT-mediated Nrf2 Signaling pathway: in vitro and in vivo studies” [Manuscript ID: IJMS 1012156] to International Journal of Molecular Science. We appreciate the time and effort that you and the reviewers have dedicated to providing valuable feedback on our manuscript. We are grateful to the reviewers for their insightful comments, which have improved the paper.

We have incorporated changes that reflect all the suggestions provided by the reviewers. All changes are highlighted in green in the revised manuscript. Please see below for our point-by point response to the reviewers’ comments. If any responses are unclear or you wish additional changes, please do not hesitate to let us know.

  1. There are number of typos were found throughout the manuscript. I advise the author should provide extra attention for those typos (Please find the attached file which I have indicated those typos.

Thank you for your valuable comments; we have made the corrections as suggested.

  1. The Figures quality must be improved.

Thank you for your valuable comments. Our revised manuscript the figure quality has been improved.

Thank you very much for all comments and questions, which allowed us to improve the quality of the manuscript.

Round 2

Reviewer 1 Report

Although the authors proceeded to some improvements of the manuscript it remains some mistakes and inaccuracies.

Experimental section

2.6

In order to process the microscopy image the DCF fluorescence intensity should be normalized with the number of cells of each image. It could be done using DAPi staining to get the number of nuclei. However, this counterstaining was not performed in that assay. Consequently, the ROS analysis is not valid.

Results

3.1

Furthermore, fig 1-3 relative to the assessment of ROS should include images of cells treated with KFL alone (which is the control for ZEA + KFL).

3.5

An image of cells exposed to NAC + ZEA is still lacking in fig 3-C: it is important to compare the effect of KFL with NAC

3-6

The results do not support an up-regulation of the Akt or PIK3 expression but an increase of the phosphorylation of those proteins. The authors did not correct the text while this point was underlined in the preceding comments.

3-8

“µmol” was still used instead of µM in Fig 5!

3.9

Appearance of cleaved PARP (fragment 85kDa) is a one of the hallmark of apoptosis: the author have to represent the level of the 85kDA fragment in a chart, rather than the complete PARP.

Author Response

Experimental section

2.6

In order to process the microscopy image the DCF fluorescence intensity should be normalized with the number of cells of each image. It could be done using DAPi staining to get the number of nuclei. However, this counterstaining was not performed in that assay. Consequently, the ROS analysis is not valid.

Thank you for your valuable comments. We are based our previous published research studies that we have used the same methods for this study (https://doi.org/10.1111/jcmm.13973 , https://doi.org/10.1016/j.bcp.2017.12.014 ). Our future study we will consider your valuable suggestion.

Results

3.1

Furthermore, fig 1-3 relative to the assessment of ROS should include images of cells treated with KFL alone (which is the control for ZEA + KFL).

 Our revised manuscript we have included KFL alone figures

3.5

An image of cells exposed to NAC + ZEA is still lacking in fig 3-C: it is important to compare the effect of KFL with NAC

Our revised manuscript we have removed KFL+ NAC+ZEA slide.

(This unwanted group, but we have included it. Thank you very much, we also thinking about this slide the previous review process.  Our previous study we never did this experiments (https://doi.org/10.1111/jcmm.13973 , https://doi.org/10.1016/j.bcp.2017.12.014 ),

3-6

The results do not support an up-regulation of the Akt or PIK3 expression but an increase of the phosphorylation of those proteins. The authors did not correct the text while this point was underlined in the preceding comments.

We have modified as suggested you (Line no: 353)

3-8

“µmol” was still used instead of µM in Fig 5!

We have changed as suggested you

3.9

Appearance of cleaved PARP (fragment 85kDa) is a one of the hallmark of apoptosis: the author have to represent the level of the 85kDA fragment in a chart, rather than the complete PARP.

We have modified as suggested you

Thank you for all your valuable comments and questions, which allowed us to improve the quality of the manuscript. If any responses are unclear or you wish additional changes, please do not hesitate to let us know.

Round 3

Reviewer 1 Report

Firstly, the authors used a fluorescent compound (H2DCF) to detect cellular ROS. They used fluorescence microscopy to record cellular fluorescence intensity but they did not get any image to show cells (or nuclei) in order to verify that variations of fluorescence intensity is due to ROS and not to the number of cells in the image. It could have be done using DAPI staining to get the number of nuclei. However, this counterstaining was not performed in that assay. Consequently, the ROS analysis is not valid.

The authors cannot argue that since a method was already published in another journal it can justify its inaccuracy in the present paper. 

Secondly, the authors intend to compare the efficacy of KFL v.s NAC to counter the oxidative effect of ZEA. Again, they used fluorescence microscopy to show DCF fluorescence. The basic principle to demonstrate that point is to show control cells treated with KFL and NAC alone and to compare the results with cells treated with KFL +ZEA and NAC + ZEA. But it is not the case even in the last version.

Consequently, either the authors bring the adding information necessary to the full demonstration of their hypothesis or they remove the part of the article dealing with ROS.

Author Response

Firstly, the authors used a fluorescent compound (H2DCF) to detect cellular ROS. They used fluorescence microscopy to record cellular fluorescence intensity but they did not get any image to show cells (or nuclei) in order to verify that variations of fluorescence intensity is due to ROS and not to the number of cells in the image. It could have be done using DAPI staining to get the number of nuclei. However, this counterstaining was not performed in that assay. Consequently, the ROS analysis is not valid.

The authors cannot argue that since a method was already published in another journal it can justify its inaccuracy in the present paper. 

We are very sorry that for missing the DAPI staining, we agree with reviewer's point. According to reviewer’s suggestion we have removed in this part from our revised manuscript

Secondly, the authors intend to compare the efficacy of KFL v.s NAC to counter the oxidative effect of ZEA. Again, they used fluorescence microscopy to show DCF fluorescence. The basic principle to demonstrate that point is to show control cells treated with KFL and NAC alone and to compare the results with cells treated with KFL +ZEA and NAC + ZEA. But it is not the case even in the last version.

Consequently, either the authors bring the adding information necessary to the full demonstration of their hypothesis or they remove the part of the article dealing with ROS.

We are very sorry that for missing the DAPI staining, we agree with reviewer's point. According to reviewer’s suggestion we have removed in this part from our revised manuscript

Round 4

Reviewer 1 Report

It remains some inaccuracies which must be corrected

3.5

The authors wrote in line 348:

 « …ZEA (40 μM) could downregulate pPI3/pAkt expression, as shown in Fig. 4A; however, after cotreatment with KFL, pPI3/pAkt  expression was significantly (p<0.05) upregulated in a dose-dependent manner.”

I previously notified in my preceding comments that the results show that ZEA and KFL act on the phosphorylation level of Akt and PI3: this mechanism does not depend on the “expression” of the protein.

The authors must rephrase the sentence to make it clear that it is phosphorylation that is involved.

In 3.6

Moreover, they wrote in3.6 “ZEA with KFL-treated cells has ab upregulated Akt”

The western blot clearly show that pAkT increased in the presence of KFL but not Akt and a chart shows that increase in Fig 1-A and 1-B. The authors must correct the sentence.

3.8

- Replace “PARP” by “cleaved PARP” in line 402

- The chart in Fig 5-C must represent cleaved PARP and not PARP

Author Response

3.5

The authors wrote in line 348:

 « …ZEA (40 μM) could downregulate pPI3/pAkt expression, as shown in Fig. 4A; however, after cotreatment with KFL, pPI3/pAkt  expression was significantly (p<0.05) upregulated in a dose-dependent manner.”

I previously notified in my preceding comments that the results show that ZEA and KFL act on the phosphorylation level of Akt and PI3: this mechanism does not depend on the “expression” of the protein.

The authors must rephrase the sentence to make it clear that it is phosphorylation that is involved.

We have rephrased the sentences to make it clear as suggested

 In 3.6

Moreover, they wrote in3.6 “ZEA with KFL-treated cells has ab upregulated Akt”

The western blot clearly show that pAkT increased in the presence of KFL but not Akt and a chart shows that increase in Fig 1-A and 1-B. The authors must correct the sentence.

The sentence has been corrected as suggested

 3.8

- Replace “PARP” by “cleaved PARP” in line 402

The word “PARP” has been replaced by “cleaved PARP” as suggested

- The chart in Fig 5-C must represent cleaved PARP and not PARP

The chart in Fig 5-C represents cleaved PARP and not PARP, as you suggested